# RLPIR: REINFORCEMENT LEARNING WITH PREFIX AND INTRINSIC REWARD

## ABSTRACT

Reinforcement Learning with Verifiable Rewards (RLVR) for large language models faces two critical limitations: (i) reliance on verifiable rewards restricts applicability to domains with accessible ground truth answers; (ii) training demands long rollouts (e.g., 16K tokens for complex math problems). We propose **R**einforcement **L**earning with **P**refix and **I**ntrinsic **R**eward (**RLPIR**), a verifier-free reinforcement learning framework that learns from intrinsic rewards while reducing compute. RLPIR includes (1) a **prefix rollout** paradigm that avoids long rollouts by optimizing only the first $L$ tokens, and (2) an **intra-group consistency reward** that eliminates reliance on verifiable rewards by measuring consistency among multiple sampled outputs. Across mathematical and general benchmarks, **RLPIR** matches RLVR's performance without ground truth, while substantially reducing training time by $6.96\times$. Moreover, **RLPIR** reduces reasoning sequence length by 45%, significantly improving the reasoning efficiency of LLMs.

## 1 INTRODUCTION

Large-scale Reinforcement Learning with Verifiable Rewards (RLVR) has demonstrated remarkable potential in advancing the reasoning capabilities of Large Language Models (LLMs), achieving breakthroughs in complex problem-solving tasks such as mathematical reasoning and code generation (Jaech et al., 2024; DeepSeek-AI et al., 2025). By leveraging external verifiers to provide precise reward signals, RLVR frameworks like GRPO (Hu et al., 2025) have enabled LLMs to refine their reasoning processes through iterative feedback.

However, RLVR faces an "impossible triangle" of practical challenges: **(1) Verifier dependence.** Reliance on domain-specific verifiers confines RLVR to domains with accessible ground-truth answers (e.g., mathematics), leaving general-domain reasoning, where answers are free-form and ambiguous, largely unexplored (Ma et al., 2025). **(2) High training cost.** The lengthy rollout sequences required for training (e.g., ~16K tokens for complex math problems) incur substantial computational overhead, limiting practical deployment (Zeng et al., 2025). **(3) Inference inefficiency.** RLVR-trained models (e.g., GRPO) tend to produce gradually longer responses during training (DeepSeek-AI et al., 2025), reducing inference efficiency.

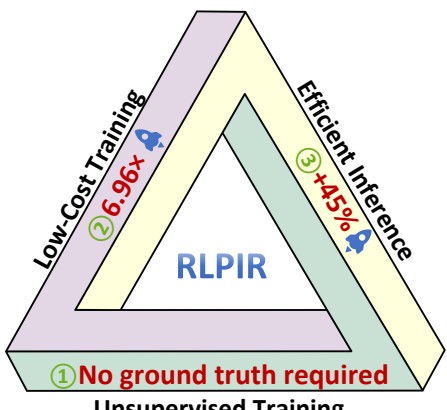

Figure 1: **RLPIR makes the "impossible triangle" possible.** It achieves (1) **unsupervised training** without ground truth, (2) **low-cost training** via prefix rollouts (~$6.96\times$ faster), and (3) **efficient inference** with a 45% reduction in reasoning length.

To address these "impossible triangle" challenges, we propose **RLPIR** (**R**einforcement **L**earning with **P**refix and **I**ntrinsic **R**eward). Our motivation is that the beginning of a solution (for example, the first 512 tokens) usually contains important decisions that determine the rest of the reasoning trajectory, yielding the correct solution. Therefore, training only on the prefix can maintain high effectiveness while increasing efficiency.

Motivated by this, RLPIR optimizes only a short prefix (e.g., $L = 512$). To enable effective training at such short lengths, RLPIR introduces two core innovations: **(1) Prefix rollouts.** A prefix rollout paradigm that truncates training sequences to $L$ tokens (e.g., 512 tokens), significantly reducing computational cost compared to RLVR baselines by focusing policy optimization on the prefix of reasoning chain, which contains critical decision points, thereby drastically reducing computational costs. **(2) Intra-group consistency reward.** For each prompt, we sample a group outputs and quantify their consistency to derive an intrinsic reward, removing the need for external verifiers.

Experimental results demonstrate that **RLPIR** matches the performance of verifier-dependent RLVR methods (e.g., GRPO) on mathematical and general benchmarks while reducing computational costs drastically without relying on ground truth answers. Notably, our framework achieves a $45\%$ reduction in reasoning sequence length, significantly improving reasoning efficiency.

Our contributions are as follows: (1) We propose **RLPIR**, a novel RL paradigm eliminating reliance on ground truth answers. (2) We develop a **prefix rollout** training strategy that reduces training time by $6.96\times$ compared to standard RLVR baselines. (3) We introduce a novel **intra-group consistency reward** that eliminates the need for external verifiers and achieves performance comparable to RLVR in mathematical domains, while demonstrating strong generalization in general domains. (4) Our method achieves a $45\%$ reduction in reasoning length during inference, achieving **efficient inference** for LLMs.

## 2 Related Work

### 2.1 Reinforcement Learning for Reasoning in LLMs

Reinforcement learning (RL) has emerged as a powerful framework for optimizing LLM reasoning, complementing supervised fine-tuning (SFT) by refining decision-making via feedback signals. Notable successes include DeepSeek-R1 (DeepSeek-AI et al., 2025) and GRPO (Hu et al., 2025), which leverage verifiable rewards such as code execution results or mathematical correctness to achieve state-of-the-art performance. However, RL with verifiable rewards (RLVR) faces two major bottlenecks: reliance on domain-specific verifiers (e.g., Math-Verify (Hynek & Greg, 2025)), which limits generality (He et al., 2025), and high computational cost from long rollouts (e.g., 16K tokens for math problems) (Zeng et al., 2025). Recent work aims to improve RL training by leveraging additional signals or trajectories. LUFFY (Yan et al., 2025) introduces off-policy guidance with high-quality reasoning trajectories and regularized importance sampling to balance imitation and exploration, outperforming pure on-policy RLVR. Token-supervised value models (Lee et al., 2025) estimate correctness probabilities at each token, enabling fine-grained credit assignment during tree search and reducing pruning errors.

### 2.2 Beyond Verifiable Rewards

To mitigate RLVR's reliance on ground-truth verifiers, researchers have explored alternative reward signals. Generative reward models (Ma et al., 2025) and self-reward mechanisms (Zhou et al., 2024) use auxiliary models or policy consistency to evaluate reasoning quality. Policy-likelihood rewards (Yu et al., 2025) extend RLVR to settings without verifiable answers but are limited to short outputs, while entropy minimization strategies (Agarwal et al., 2025) encourage deterministic reasoning at the risk of suppressing diversity. Another promising direction leverages internal consistency signals: Xie et al. (2024) decode intermediate layer predictions and weight self-consistent reasoning paths to improve calibration in chain-of-thought reasoning. Our intra-group consistency reward generalizes this idea to group-level semantic similarity, providing a differentiable intrinsic reward that eliminates reliance on external verifiers.

### 2.3 Efficient Reinforcement Learning Training Paradigms

Efficiency remains a key challenge in RL for reasoning. Full-rollout RLVR is computationally expensive, motivating research on more efficient paradigms. TTRL (Zuo et al., 2025) and Absolute Zero (Zhao et al., 2025) explore test-time refinement and self-play but remain task-specific. Controlled decoding approaches such as prefix scorers (Mudgal et al., 2024) bias generation under a reward–KL tradeoff to reduce inference-time cost. Prefix-based training has also been studied for

supervised setups (Ji et al., 2025) but its integration into RL remains underexplored. Our method addresses this gap by using prefix rollouts, truncating training to critical decision points and reducing compute by $6.96\times$ relative to standard RLVR.

## 3 PRELIMINARY

### 3.1 REINFORCEMENT LEARNING WITH VERIFIABLE REWARDS (RLVR)

Reinforcement Learning with Verifiable Rewards (RLVR) trains large language models (LLMs) using programmatically verifiable signals of correctness—such as mathematical validity, logical consistency, or code execution results.

Given an input $q$ and an output trajectory $\tau$, RLVR defines a verifiable reward $r(q, \tau) \in \mathbb{R}$ that quantifies the correctness of $\tau$ based on predefined rules or external verification tools. For example, in code generation, $r(q, \tau)$ can indicate whether the generated program passes a suite of unit tests. With a policy model $\pi_\theta$, we sample a response $\tau \sim \pi_\theta(\cdot \mid q)$ and compute the reward as

$$r(q, \tau) = \mathbb{I}[\text{Verify}(q, \tau) = \text{True}], \tag{1}$$

where $\text{Verify}(\cdot)$ is a deterministic function that checks whether $\tau$ meets the specified correctness criteria. The RLVR objective maximizes the expected verifiable reward while regularizing the policy towards a reference model $\pi_{\text{ref}}$:

$$\max_{\pi_\theta} \ \mathbb{E}_{\tau \sim \pi_\theta(\cdot \mid q)}[\, r(q, \tau)\,] \ - \ \beta\, \mathbb{D}_{\text{KL}}\big(\pi_\theta \,\|\, \pi_{\text{ref}}\big), \tag{2}$$

where $\beta > 0$ controls the strength of the regularization.

RLVR is commonly instantiated with policy-gradient methods such as REINFORCE (Williams, 1992), PPO (Schulman et al., 2017), or GRPO (Shao et al., 2024a). The GRPO objective (omitting clipping for brevity) is

$$\mathcal{J}_{\text{GRPO}}(\theta) = \mathbb{E}_{\{\tau_g\}_{g=1}^{G} \sim \pi_{\theta_{\text{ref}}}(\cdot \mid q)} \frac{1}{G} \sum_{g=1}^{G} \left( \frac{\pi_\theta(\tau_g \mid q)}{\pi_{\theta_{\text{ref}}}(\tau_g \mid q)} A_g - \beta\, \mathbb{D}_{\text{KL}}(\pi_\theta \,\|\, \pi_{\text{ref}}) \right) \tag{3}$$

where the KL-divergence term is calculated as:

$$\mathbb{D}_{\text{KL}}\big(\pi_\theta \,\|\, \pi_{\text{ref}}\big) = \frac{\pi_{\text{ref}}(\tau_g \mid q)}{\pi_\theta(\tau_g \mid q)} - \log \frac{\pi_{\text{ref}}(\tau_g \mid q)}{\pi_\theta(\tau_g \mid q)} - 1, \tag{4}$$

and the advantage $A_g$ is computed via group-wise standardization of rewards $\{r_1, \ldots, r_G\}$:

$$A_g \ = \ \frac{r_g - \text{mean}(\{r_1, \ldots, r_G\})}{\text{std}(\{r_1, \ldots, r_G\})}. \tag{5}$$

Despite its appeal, RLVR still hinges on ground truth: rewards exist only when a trusted verifier or gold answer is available. Moreover, $r(q, \tau)$ is computable only after the model completes a full trajectory, yielding delayed and often sparse feedback that is costly and domain-specific (Liu et al., 2025; Team et al., 2025). These properties limit its applicability beyond well-specified domains and tasks with ambiguity or subjective goals.

## 4 MOTIVATION

This work proposes a paradigm shift from full reasoning trajectory optimization with verifiable rewards to prefix optimization without relying on verifiable rewards, aiming at addressing crucial challenges for RLVR. This section presents the preliminary studies that ground the motivations of this proposal, and provide evidence supporting the design of **prefix rollout** and **intra-group consistency reward**, the two core elements of RLPIR.

## 4.1 Prefix Optimization Suffices to Improve Reasoning

The first study explores whether **prefix optimization can play a similar role to full-length trajectory optimization for RL training**. For this purpose, we fine-tuned Qwen3-0.6B with DPO on the AIME24 benchmark under the full-length reasoning trajectory (i.e., Full-length DPO) and 512-token prefix (i.e., Prefix DPO) settings. As shown in Table 1, prefix-only DPO substantially improves over the Qwen3-0.6B base model and performs nearly on par with full-length DPO (See Section E.2 for the training reward dynamics). This suggests that the first 512 tokens capture most of the learnable signal and that the policy learned on short prefixes generalizes to full-length reasoning trajectories. **It motivates us to adopt prefix optimization in the RLPIR method.**

| Model | AIME24 Accuracy (%) |
|---|---|
| Qwen3-0.6B | 9.67 |
| + Full-length DPO | 13.7 |
| + Prefix DPO (512 tokens) | **13.3** |

Table 1: Prefix DPO versus Full-length DPO on AIME24.

## 4.2 High-consistency Prefix Yields High-quality Reasoning Trajectory

The second study investigates the strategy for prefix optimization in the absence of verifiable rewards. We hypothesize that within a group of decent-looking reasoning trajectories, **the prefix that is most semantically related to others is most likely to yield a correct reasoning trajectory.** To test this hypothesis, we designed a controlled "forced-prefix continuation" task, where we forced **Qwen3-8B** to generate the full chain-of-thought solution based on the prefix on the **AIME24** benchmark. For each problem, 64 full chain-of-thought solutions were sampled as marked as correct/incorrect based on their final answers. Then, the "best" prefix was selected with three different strategies from the first $L$ tokens of the solutions:

- **A: High-consistency.** The first $L$ tokens of all solutions ($K = 64$ in total) were embedded[1]. Then, for each solution $i$ with embedding $e_i$, we estimated its overall semantically relatedness with other prefixes by computing its **intra-group consistency score** $c_i = \frac{1}{K} \sum_i^K \cos(e_i, e_j)$. The prefix with the greatest value of $c_i$ was selected.

- **B: Random correct.** The best prefix was randomly picked from the correct solutions with a uniform distribution.

- **C: Random incorrect.** The best prefix was randomly picked from the incorrect solutions with a uniform distribution.

| Prefix Length | Prefix Selection Strategy | Accuracy (%) |
|---|---|---|
| | A: High-consistency | **77.1** |
| 512 tokens | B: Random correct | 75.0 |
| | C: Random incorrect | 51.4 |
| | A: High-consistency | **78.1** |
| 1024 tokens | B: Random correct | 77.2 |
| | C: Random incorrect | 50.1 |
| *Qwen3-8B baseline* | | 73.0 |

Table 2: Accuracy of forced-prefix continuation under different prefix selection strategies on AIME24. The high-consistency prefixes markedly boost performance over random correct ones, while incorrect-solution prefixes cause severe degradation.

Table 2 shows the accuracy of the forced-prefix continuation task on AIME24, where the prefixes were selected with all three strategies above with length $L$=512 or 1024. At both lengths, the high-consistency prefixes attain a superior performance over random correct, with a more remarkable gap at $L$=512 where high-consistency prefixes raises accuracy to **77.1%** (vs 73.0% baseline). These results indicate that the first few hundred tokens capture the pivotal decisions of the reasoning chain, justifying our training strategy of **intra-group consistency reward**, where we generalize this idea into a differentiable intrinsic reward for reinforcement learning.

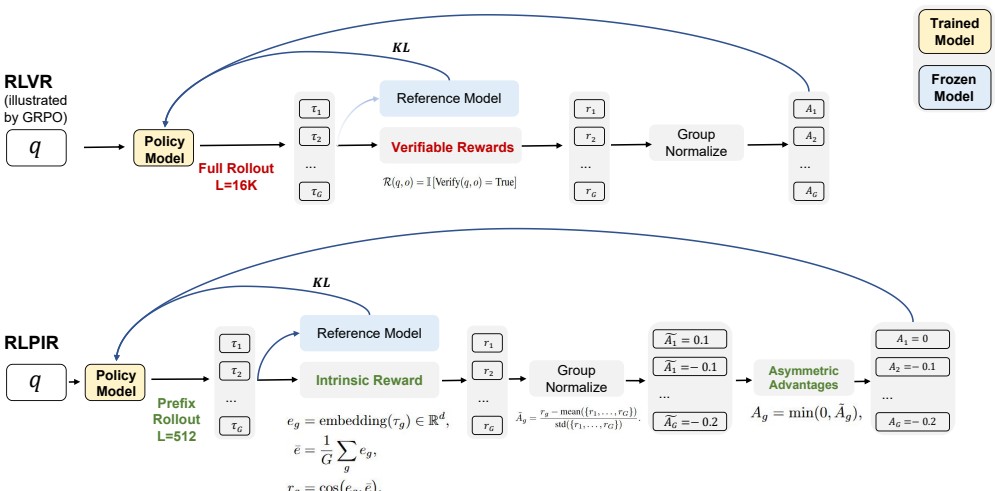

Figure 2: Comparison of RLVR (**R**einforcement **L**earning with **V**erifiable **R**eward) and **RLPIR** (**R**einforcement **L**earning with **P**refix and **I**ntrinsic **R**eward). While RLVR suffers from high computational costs due to long rollouts and relies on ground-truth verification, **RLPIR** achieves efficient training via short prefix rollouts and intrinsic rewards without requiring external verification. In addition, RLPIR employs **Asymmetric Advantages** to prevent reward hacking and maintain diversity while regularizing the policy effectively.

## 5 METHODOLOGY

As motivated in Section 4, we introduce **R**einforcement **L**earning with **P**refix and **I**ntrinsic **R**eward (**RLPIR**), a novel reinforcement learning framework that addresses the limitations of traditional RLVR by introducing two key innovations: (1) a **prefix rollout** paradigm that optimizes only the first $L$ tokens, and (2) an **intra-group consistency reward** that measures consistency among multiple sampled outputs, eliminating reliance on verifiable rewards. Figure 2 illustrates the framework's architecture compared to conventional RLVR (e.g., GRPO) approaches.

### 5.1 PROBLEM FORMULATION

Let $q$ denote an input prompt drawn from a dataset $\mathcal{D}$. The policy model $\pi_\theta$ produces a partial trajectory $\tau = (t_1, \ldots, t_L)$ consisting of the first $L$ tokens of the full reasoning chain. Our goal is to maximize the expected intrinsic reward $r(q, \tau)$ while constraining policy drift with a KL-divergence penalty:

$$\max_\theta \ \mathbb{E}_{q\sim\mathcal{D},\, \tau\sim\pi_\theta(\cdot|q)}\Big[r(q,\tau) - \beta\, \mathbb{D}_{\mathrm{KL}}\big(\pi_\theta \| \pi_{\mathrm{ref}}\big)\Big], \tag{6}$$

where $\pi_{\mathrm{ref}}$ is the frozen reference policy and $\beta$ controls the regularisation strength. The remainder of this section details the design of $r(q, \tau)$ and the prefix rollout schedule.

### 5.2 PREFIX ROLLOUT

Long rollouts (e.g., $\sim$ 16K tokens math) dominate the wall-clock cost in Reinforcement Learning with Verifiable Rewards (RLVR). Inspired by the analysis in Section 4, we train exclusively on the initial prefix of length $L = 512$ tokens. During training, rewards and policy gradients are computed only over this prefix of $L$ tokens. However, at evaluation time, the model is allowed to generate freely beyond $L$ tokens to complete the output.

---

[1]We use all-MiniLM-L6-v2 as embedding model

### 5.3 INTRA-GROUP CONSISTENCY REWARD

We achieve the **intrinsic reward** via **Intra-group Consistency**, using semantic similarity as the reward signal. For each prompt $q$ we sample $G$ independent rollouts $\{\tau_g\}_{g=1}^G$ under the current policy. Each rollout $\tau_g$ is embedded with a sentence encoder[2]. We measure similarity to the group center $\bar{e}$ using cosine similarity, as described in Section 4.2 for computing high consistency:

$$e_g = \text{embedding}(\tau_g) \in \mathbb{R}^d, \qquad \bar{e} = \frac{1}{G}\sum_g e_g, \qquad r_g = \cos(e_g, \bar{e}). \tag{7}$$

### 5.4 ASYMMETRIC ADVANTAGES

To mitigate reward hacking caused by excessive similarity, which could lead to model collapse, we adopt an asymmetric advantage mechanism. We first compute the cosine similarity scores $r_g$, standardize them, and then convert the standardized scores into asymmetric advantages by clipping the positive branch:

$$\tilde{A}_g = \frac{r_g - \text{mean}(\{r_1, \dots, r_G\})}{\text{std}(\{r_1, \dots, r_G\})}, \qquad A_g = \min(0, \tilde{A}_g). \tag{8}$$

Only prefixes that are less consistent than the group average ($\tilde{A}_g < 0$) get a non-zero (negative) advantage. We penalize those low-consistency samples and give no reward to already-similar ones. This prevents reward hacking while keeping useful diversity.

## 6 EXPERIMENTS

### 6.1 TRAINING SETUP

We implement **RLPIR** using Nemo-RL (nem, 2025). For each problem in a training batch, we generate a group of $G = 16$ candidate solutions. Crucially, each rollout is limited to $L = 512$ tokens, as motivated by the ablation results in Section 8.1.

The intrinsic reward for each prefix is calculated based on its semantic consistency within its group. We embed each $L = 512$ token prefix using the all-MiniLM-L6-v2[3] sentence encoder.

For policy optimization, we use GRPO (Shao et al., 2024b). We process a batch of 32 problems per step, with a constant KL-divergence penalty of $\beta = 0.001$ to regularize the policy and prevent deviation from the reference model. All models are trained using the AdamW optimizer with a learning rate of $1 \times 10^{-6}$. Experiments were conducted on $8 \times$ NVIDIA A100 GPUs.

### 6.2 MODELS AND TRAINING DATA

Our experiments are conducted on several base models to demonstrate the broad applicability of RLPIR. We apply our training method to Llama (Meta AI, 2024), Qwen2.5 (Yang et al., 2024) and Qwen3 series (Team, 2025a).

We construct our training set from two public math corpora, OpenR1-Math-220k[4] and Big-Math-RL-Verified[5]. For each problem, we run inference with three models (Deepseek R1 1.5B, Deepseek R1 7B (DeepSeek-AI et al., 2025), and QWQ 32B (Team, 2025b)) and log their correctness. We then define a four-stage data split strategy: problems solved by the 1.5B model are labeled as Level 1 (easiest); those missed by 1.5B but solved by 7B form Level 2; items only solved by the 32B model become Level 3; and those unsolved by all three are Level 4 (hardest). This pipeline filters out trivially easy or completely intractable items, yielding a challenging yet learnable dataset focused

---

[2]We use `all-MiniLM-L6-v2` as the embedding model.

[3]https://huggingface.co/sentence-transformers/all-MiniLM-L6-v2

[4]https://huggingface.co/datasets/open-r1/OpenR1-Math-220k

[5]https://huggingface.co/datasets/SynthLabsAI/Big-Math-RL-Verified

| Model | General | | | Math | | | | Avg | |
|---|---|---|---|---|---|---|---|---|---|
| | MMLU-Pro ↑ | GPQA ↑ | SuperGPQA ↑ | AIME 24 ↑ | AIME 25 ↑ | Olympiad ↑ | Minerva ↑ | General ↑ | Math ↑ |
| | | | *Llama Models* | | | | | | |
| Llama3.1-8B-Inst | 46.9 | 30.2 | **22.2** | 3.0 | 0.0 | 13.0 | 10.2 | 33.1 | 6.6 |
| **+RLPIR(ours)** | **47.0** | **31.8** | 21.0 | **4.3** | 0.0 | **15.3** | **12.6** | **33.2** | **8.1** |
| | | | *Qwen2.5 Models* | | | | | | |
| Qwen2.5-7B-Inst | 56.6 | 33.8 | 29.0 | 11.6 | 8.5 | 34.2 | 26.1 | 39.8 | 20.1 |
| **+RLPIR(ours)** | **58.5** | **35.5** | **31.6** | **16.2** | **14.7** | **38.3** | **31.1** | **41.9** | **25.1** |
| Qwen2.5-14B-Inst | 62.7 | 41.4 | 35.0 | 11.3 | 11.0 | 37.3 | 29.7 | 46.4 | 22.3 |
| **+RLPIR(ours)** | **65.5** | **42.9** | **38.3** | **16.2** | **15.4** | **42.4** | **34.0** | **48.9** | **27.0** |
| | | | *Qwen3 Models* | | | | | | |
| Qwen3-4B-Inst | 63.7 | 53.0 | **42.4** | 72.6 | 64.3 | 61.4 | 33.4 | 53.0 | 57.9 |
| +RLVR | 59.5 | 50.4 | 33.8 | **80.9** | **70.7** | 66.5 | **43.2** | 47.9 | **65.3** |
| **+RLPIR(ours)** | **65.1** | **53.9** | 42.0 | 77.3 | 69.8 | 65.7 | 38.6 | **53.7** | 62.9 |
| Qwen3-8B-Inst | 67.7 | 61.1 | **48.5** | 73.0 | 66.0 | 63.5 | 35.6 | 59.1 | 59.5 |
| +RLVR | 65.8 | 61.7 | 40.6 | **80.1** | **73.3** | 68.5 | **40.8** | 56.0 | **65.7** |
| **+RLPIR(ours)** | **69.7** | **62.2** | 46.2 | 78.8 | 72.2 | **69.3** | 40.0 | **59.3** | 65.1 |
| Qwen3-14B-Inst | 72.4 | 65.1 | 52.5 | 80.0 | 70.3 | 63.4 | 37.1 | 63.3 | 62.7 |
| +RLVR | 71.5 | 61.0 | 50.3 | **86.6** | **78.7** | 66.1 | 40.2 | 60.9 | 67.9 |
| **+RLPIR(ours)** | **75.1** | **66.4** | **53.0** | 86.2 | 76.9 | **67.9** | **41.9** | **64.8** | **68.2** |

Table 3: **Main results.** RLVR is implemented as full-length GRPO$_{16K}$; RLPIR uses a 512-token prefix during training. **Without any external verifiers or ground-truth labels**, **RLPIR** attains math performance on par with verifier-dependent RLVR baselines. Beyond mathematics, **RLPIR** is **consistently more robust on general-domain benchmark**

on informative examples. After filtering, our splits contain 154,817, 80,486, 25,309, and 74,825 problems for Levels 1–4 respectively. Unless noted, all experiments are trained on the Level 3 data. Moreover, the correct/incorrect sample pairs obtained in this process constitute the DPO training data used for the training in Section 4.1.

## 6.3 BASELINES

We compare the performance of RLPIR against several strong baselines to comprehensively evaluate its effectiveness. Importantly, we do not include comparisons with verifier-free methods based on probabilistic rewards such as RLPR (Yu et al., 2025), since our method is explicitly designed for settings without any ground truth, whereas those approaches still rely on ground-truth signals for evaluation or reward construction.

**Base models.** We report the baseline performance of models without any further training, including Llama (Meta AI, 2024), Qwen2.5 (Yang et al., 2024), and the Qwen3 series (Team, 2025a), to verify the effectiveness of our method across different models.

**RL with Verifiable Rewards (RLVR).** As a verifier-dependent baseline, we implement GRPO with full-length rollouts (GRPO$_{16K}$), where rewards are computed by programmatic verification against ground-truth answers. Optimizing on full trajectories (up to ∼16K tokens) yields substantially higher cost but represents an approximate upper bound in performance, serving as a reference for our more efficient verifier-free approach. RLVR uses the same training data in Section 6.2.

## 6.4 EVALUATION

We evaluate the reasoning capabilities with multiple general reasoning and mathematical benchmarks. For math reasoning, we include Olympiad, Minerva, AIME24 and AIME25. For general domains, we include MMLU-Pro (Wang et al., 2024), GPQA (Rein et al., 2023), and SuperGPQA.

## 7 MAIN RESULTS

### 7.1 TRAINING EFFECTIVENESS AND GENERALIZATION

Table 3 shows that, **without any external verifiers or ground-truth labels**, **RLPIR** attains math performance on par with verifier-dependent RLVR baselines (e.g., GRPO$_{16K}$) while optimizing only a 512-token prefix. This indicates that high-fidelity reasoning signals can be learned from intrinsic, intra-group consistency alone. Beyond mathematics, **RLPIR** is **consistently more robust on**

**general-domain benchmarks** (e.g., MMLU-Pro, GPQA) than both the base models and RLVR, underscoring stronger cross-task transfer when no domain-specific verifiers are available. Taken together, these results demonstrate that RLPIR matches RLVR in verifier-friendly domains and surpasses it in verifier-scarce domains, while also delivering substantially lower training cost via short-prefix rollouts.

## 7.2 COMPUTE EFFICIENCY

We benchmark wall-clock training time on Qwen3-8B for 1000 optimization steps using identical hardware and hyper-parameters. RLVR ($GRPO_{16K}$) requires **177.5 hours**, whereas **RLPIR** completes in **25.5 hours**,

| Method | Time | Time/step | Speed-up |
|---|---|---|---|
| RLVR ($GRPO_{16K}$) | 177.5 h | 10.65 min | - |
| **RLPIR** (ours, $L = 512$) | 25.5 h | 1.53 min | **6.96×** |

Table 4: Compute Efficiency. Wall-clock training time on **Qwen3-8B** ($8 \times$ A100), 1000 steps.

yielding a **6.96×** speed-up and an **85.6%** reduction in wall-clock time. The observed gains are consistent with our rollout budget: RLPIR trains on **512-token** prefixes while RLVR consumes $\sim$ **16K** tokens per step. RLPIR substantially lowers training cost.

## 7.3 REASONING EFFICIENCY

We measure reasoning efficiency by the average number of tokens generated on the AIME24 benchmark. Table 5 compares the average response lengths of Qwen3 models in three scenarios: the original base model, after RLVR training, and after RLPIR training. While RLVR-based optimization tends to increase response length, **RLPIR** produces markedly shorter response while maintaining accuracy. See Section I.1 for a detailed case.

| Setting | Qwen3-4B (tokens) | Qwen3-8B (tokens) | Qwen3-14B (tokens) |
|---|---|---|---|
| Base | 14229 | 14539 | 15280 |
| +RLVR ($GRPO_{16K}$) | 15846 | 16483 | 17294 |
| **+RLPIR** (ours, $L = 512$) | **11772** | **9564** | **9474** |
| $\Delta$ vs Base | −17.3% | −34.2% | −38.0% |
| $\Delta$ vs RLVR | −25.7% | −42.0% | −45.2% |

Table 5: Reasoning efficiency. Average response lengths (tokens) on AIME24 by model (columns). Percent changes are computed for RLPIR relative to the indicated baseline.

| Prefix Len (tokens) | Acc↑ | $\Delta$Acc | Len↓ (tokens) | $\Delta$Len(%) |
|---|---|---|---|---|
| Qwen3-8B | 73.0 | - | 14539 | - |
| 256 | 76.3 | +3.3 | 9866 | -32.2% |
| 512 | **78.8** | **+5.8** | 11797 | -18.9% |
| 1024 | 77.0 | +4.0 | 14601 | +0.4% |

Table 6: Effect of prefix length $L$ with Qwen3-8B on AIME24. A 512-token prefix achieves the best accuracy while still shortening solutions substantially.

# 8 ABLATION STUDY

## 8.1 EFFECT OF PREFIX LENGTH

We ablate the rollout prefix budget $L$ to understand its effect on both final accuracy and reasoning efficiency. Using the **Qwen3-8B** backbone and **AIME24** as the validation set, we train **RLPIR** with three prefix lengths ($L \in \{256, 512, 1024\}$) under identical settings (Section 6.2) and report accuracy as well as the average response length at evaluation time (the model is free to generate beyond the prefix length at test time).

Table 6 shows that a **512-token** prefix yields the best accuracy, while still providing substantial length reduction relative to the base model. Shorter prefixes (*e.g.*, $L = 256$) further compress response but slightly underperform in accuracy, suggesting that the reward signal becomes less informative when too little of the early reasoning is observed. Conversely, longer prefixes ($L = 1024$) do not improve accuracy and even increase average length back to the base level.

## 8.2 EFFECT OF ASYMMETRIC ADVANTAGES

We study the impact of **Asymmetric Advantages** in Eq. equation 8, which penalize only low-consistency rollouts and assign zero advantage to highly consistent ones. This design aims to deter reward hacking behaviors (such as trivial repetition) that can artificially inflate similarity.

As shown in Table 7, removing the Asymmetric clipping causes severe degradation: accuracy collapses to 42.3% and outputs become extremely short (average length $\sim$6.5K tokens), consistent with a mode-seeking failure where the policy inflates similarity by emitting degenerate continuations. By contrast, **RLPIR** with Asymmetric advantages attains higher accuracy and shorter outputs versus the base model, indicating that the clipped signal effectively regularizes the policy away from collapse while preserving legitimate diversity. See Section I.3 for an illustration.

## 8.3 EFFECT OF TRAINING-DATA DIFFICULTY

We study how the difficulty of training data affects both accuracy and reasoning efficiency. Recall from Section 6.2 that we partition the training dataset into four levels (Level 1–4, easiest→hardest) using success rates from progressively stronger solvers. Using Qwen3-8B as the backbone, we fine-tune with **RLPIR** on each single level separately and evaluate on AIME24. See table 8 for details.

**Findings. (1) Accuracy peaks at medium difficulty.** All levels improve accuracy over the base model, with the best score attained by Level 3 (hard-but-solvable) and a mild drop at Level 4. We hypothesize that the intrinsic consistency signal benefits from items that are challenging enough to elicit diverse prefixes but not so hard that group rollouts become uniformly noisy. Overly easy items (Level 1) provide limited gradient signal because most prefixes already agree; overly difficult items (Level 4) increase variance and reduce the reliability of the group-consistency reward. **(2)**

| Setting | Acc↑ | ΔAcc | Len↓ (tokens) | ΔLen (%) |
|---|---|---|---|---|
| Qwen3-8B | 73.0 | 0.0 | 14539 | - |
| RLPIR (L=512) | **78.8** | +5.8 | 11797 | -18.9% |
| w/o Asymmetric Advantages | 42.3 | -30.7 | **6543** | -55.0% |

Table 7: Effect of Asymmetric advantages (Eq. 8) on Qwen3-8B (AIME24). Removing it leads to reward hacking and large accuracy drops despite shorter outputs.

| Setting | Acc↑ | Δ Acc vs Base | Len↓ (tokens) | Δ Len(%) vs Base |
|---|---|---|---|---|
| Qwen3-8B | 73.0 | - | 14539 | - |
| Level 1 (easy) | 77.3 | +4.3 | 13007 | −10.5% |
| Level 2 | 78.0 | +5.0 | 12423 | −14.6% |
| Level 3 | 78.8 | +5.8 | 11797 | −18.9% |
| Level 4 (hard) | 77.6 | +4.6 | 9564 | −34.2% |
| Random | 78.1 | +5.1 | 12073 | −16.9% |

Table 8: Effect of training-data difficulty on AIME24 with Qwen3-8B. "Random" uses 20,000 examples randomly sampled from the entire dataset.

**Reasoning efficiency improves the most with higher difficulty.** Average solution length decreases monotonically as difficulty increases, with the largest compression observed at Level 4 (Table 8). Harder items induce stronger disagreement across sampled prefixes, which yields larger Asymmetric penalties (Eq. 8) on off-manifold trajectories. Under **RLPIR**, the policy therefore learns to commit earlier to high-consistency paths, pruning meandering continuations and producing shorter final chains despite training only on 512-token prefixes. **(3) Robustness to training-data difficulty.** RLPIR is robust to the difficulty distribution of training data: training on a randomly sampled subset of the full dataset yields gains comparable to difficulty-stratified subsets (Table 8).

## 9 CONCLUSION AND FUTURE WORK

In this work we introduced **RLPIR**, Reinforcement Learning with Prefix and Intrinsic Reward, a verifier free training paradigm that allows large language models to attain the "impossible trinity" by simultaneously achieving: (1) **unsupervised training** without ground truth, (2) **low-cost training** via prefix training, yielding a $6.96\times$ speedup in training, and (3) **efficient inference** with a **45%** reduction in reasoning length. In contrast, traditional RLVR methods (e.g., GRPO) rely on external verifiers and full-length rollouts ($\sim$16K-token in math), which are costly and typically result in longer responses during inference. RLPIR achieves these goals through two key innovations: (a) a **prefix rollout** paradigm that optimizes only the first $L$ tokens, (b) an **intra-group consistency reward** that measures consistency among multiple sampled outputs, eliminating reliance on verifiable rewards. Across mathematical and general benchmarks, **RLPIR** matches RLVR's (e.g., GRPO) performance without ground truth, while substantially lowering training time by $6.96\times$. Moreover, our method reduces reasoning sequence length by 45%, significantly improving the reasoning efficiency of LLMs. Moreover, RLPIR exhibits superior domain generalization compared to RLVR, as its verifier-free design avoids overfitting to narrow, task-specific reward signals, enabling robust transfer across diverse and open-ended domains. Future work will focus on extending RLPIR to additional domains and models to further validate its scalability and generalization capabilities.

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

## A  ETHICS STATEMENT

This work adheres to the ICLR Code of Ethics. In this study, no human subjects or animal experimentation was involved. All datasets used, were sourced in compliance with relevant usage guidelines, ensuring no violation of privacy. We have taken care to avoid any biases or discriminatory outcomes in our research process. No personally identifiable information was used, and no experiments were conducted that could raise privacy or security concerns. We are committed to maintaining transparency and integrity throughout the research process.

## B  REPRODUCIBILITY STATEMENT

We have made every effort to ensure that the results presented in this paper are reproducible. All code and datasets have been made publicly available in an anonymous repository to facilitate replication and verification. The experimental setup, including training steps, model configurations, and hardware details, is described in detail in the paper. We expect these practices will help the community validate our work and push forward future advancements in the field.

## C  LLM USAGE

Large Language Models (LLMs) were used to aid in the writing and polishing of the manuscript. Specifically, we used an LLM to assist in refining the language, improving readability, and ensuring clarity in various sections of the paper. The model helped with tasks such as sentence rephrasing, grammar checking, and enhancing the overall flow of the text.

It is important to note that the LLM was not involved in the ideation, research methodology, or experimental design. All research concepts, ideas, and analyses were developed and conducted by the authors. The contributions of the LLM were solely focused on improving the linguistic quality of the paper, with no involvement in the scientific content or data analysis.

The authors take full responsibility for the content of the manuscript, including any text generated or polished by the LLM. We have ensured that the LLM-generated text adheres to ethical guidelines and does not contribute to plagiarism or scientific misconduct.

## D  LIMITATIONS

RLPIR has several limitations: (i) while we empirically validate the effectiveness of a fixed prefix length $L = 512$, the framework does not yet include an adaptive mechanism for selecting $L$, which could further improve robustness across diverse tasks and contexts; (ii) as is common in reinforcement learning methods, performance can be sensitive to hyperparameters such as KL weight, prefix length, and learning rate; (iii) due to hardware resource constraints, our experiments focus on representative domains and model sizes, so broader validation remains an important direction for future work.

## E  DATA PREPARATION

### E.1  TRAINING DATA DISTRIBUTION

We construct our training set from two public math corpora, OpenR1-Math-220k[6] and Big-Math-RL-Verified[7]. For each problem, we run inference with three models (Deepseek R1 1.5B, Deepseek R1 7B (DeepSeek-AI et al., 2025), and QWQ 32B (Team, 2025b)) and log their correctness. We then define a four-stage data split strategy: problems solved by the 1.5B model are labeled as Level 1 (easiest); those missed by 1.5B but solved by 7B form Level 2; items only solved by the 32B model become Level 3; and those unsolved by all three are Level 4 (hardest). This pipeline filters out trivially easy or completely intractable items, yielding a challenging yet learnable dataset focused

---

[6]https://huggingface.co/datasets/open-r1/OpenR1-Math-220k
[7]https://huggingface.co/datasets/SynthLabsAI/Big-Math-RL-Verified

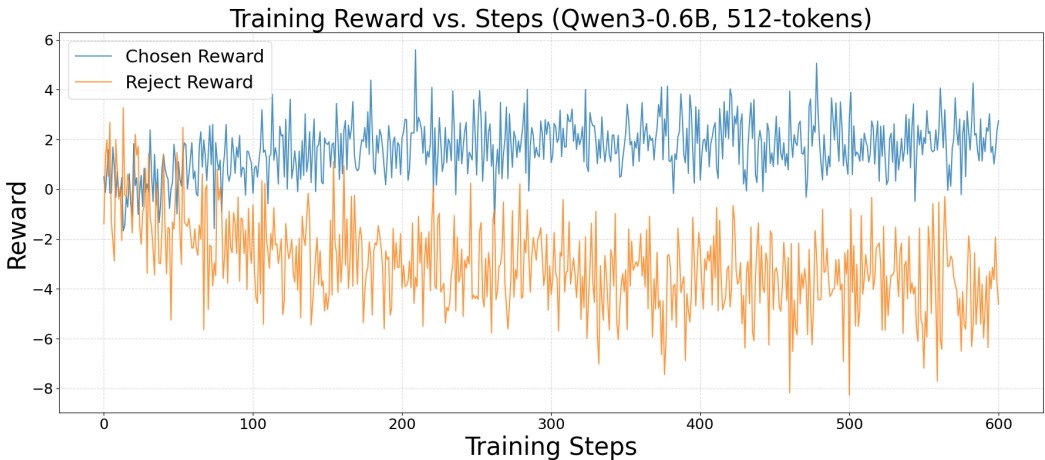

Figure 3: **Prefix-only DPO on Qwen3-0.6B** ($L$=512). The reward for chosen prefixes (blue) rises while the reward for rejected prefixes (orange) falls. This supports our claim that the first $L$ tokens contain sufficient information to learn a robust reasoning policy, motivating RLPIR's short-prefix rollouts.

on informative examples. After filtering, our splits contain 154,817, 80,486, 25,309, and 74,825 problems for Levels 1–4 respectively. Unless noted, all experiments are trained on the Level 3 data.

| Level | # Problems | Proportion (%) |
|---|---|---|
| 1 (easiest) | 154,817 | 46.15 |
| 2 | 80,486 | 23.99 |
| 3 | 25,309 | 7.55 |
| 4 (hardest) | 74,825 | 22.31 |
| **Total** | 335,437 | 100.00 |

Table 9: Curriculum levels and problem counts in our training set.

### E.2 PREFIX DPO TRAINING

**Setup.** We use the same training corpus as in the main experiments. During corpus curation we additionally obtain positive/negative pairs per problem (correct vs. incorrect solutions). From each problem we construct a 512-token prefix pair.

**Training.** We fine-tune **Qwen3-0.6B** with DPO on these prefix pairs.

**Results.** Rewards for chosen prefixes increase while those for rejected prefixes decrease, indicating that prefixes alone provide a learnable signal. This supports our claim that the first $L$ tokens contain sufficient information to learn a robust reasoning policy, motivating RLPIR's short-prefix rollouts.

### E.3 INITIAL DATA COLLECTION

Our training data is sourced from two high-quality mathematical reasoning datasets: SynthLabsAI/Big-Math-RL-Verified and open-r1/OpenR1-Math-220k. The initial data collection process involves downloading and preprocessing these datasets to create a unified training corpus.

```python
import os
import jsonlines
from datasets import load_dataset
from random import shuffle

dataset_name_1 = "SynthLabsAI/Big-Math-RL-Verified"
dataset_name_2 = "open-r1/OpenR1-Math-220k"
```

```
8  dataset_name_list = [dataset_name_1, dataset_name_2]
9
10 def process_prompt(example):
11     messages = [
12         {
13             "role": "system",
14             "content": "You are a helpful and harmless assistant. You
       should think step-by-step.",
15         },
16         {
17             "role": "user",
18             "content": example["problem"]
19         },
20     ]
21     return {"messages": messages}
22
23 def preprocess_dataset(dataset_name):
24     dataset = load_dataset(dataset_name, split="train")
25     dataset = dataset.map(process_prompt, num_proc=64, batched=False)
26     return [x["messages"] for x in dataset], [x["answer"] for x in
       dataset], [x["problem"] for x in dataset]
27
28 # Collect and deduplicate data
29 messages_list, answer_list, problem_list = [], [], []
30 for dataset_name in dataset_name_list:
31     m, a, p = preprocess_dataset(dataset_name)
32     messages_list.extend(m)
33     answer_list.extend(a)
34     problem_list.extend(p)
35
36 # Remove duplicates based on problem content
37 messages_list, answer_list, problem_list = unique_list(messages_list,
       answer_list, problem_list)
```

Listing 1: Initial Data Collection Script

### E.4 DIFFICULTY-BASED DATA CATEGORIZATION

We implement a novel difficulty-based categorization system using three reference models of varying capabilities: DeepSeek-R1-Distill-Qwen-1.5B (1.5B parameters), DeepSeek-R1-Distill-Qwen-7B (7B parameters), and QwQ-32B (32B parameters). Each problem is evaluated by all three models, and the difficulty level is determined based on the models' success rates.

---

**Algorithm 1** Difficulty-Based Data Categorization

---

**Require:** Problems $P$, Models $M = \{M_{1.5B}, M_{7B}, M_{32B}\}$
**Ensure:** Difficulty levels $L = \{L_1, L_2, L_3, L_4\}$
1: **for** each problem $p \in P$ **do**
2:     $c_{1.5B} \leftarrow \text{Evaluate}(p, M_{1.5B})$
3:     $c_{7B} \leftarrow \text{Evaluate}(p, M_{7B})$
4:     $c_{32B} \leftarrow \text{Evaluate}(p, M_{32B})$
5:     **if** $c_{1.5B} = 1$ **then**
6:         Assign $p$ to $L_1$ (Easiest)
7:     **else if** $c_{7B} = 1$ **then**
8:         Assign $p$ to $L_2$
9:     **else if** $c_{32B} = 1$ **then**
10:       Assign $p$ to $L_3$
11:     **else**
12:       Assign $p$ to $L_4$ (Hardest)
13:     **end if**
14: **end for**

---

The categorization results in four distinct difficulty levels:

- **Level 1 (Easiest)**: Problems solved correctly by the 1.5B model
- **Level 2**: Problems failed by 1.5B but solved by 7B model
- **Level 3**: Problems failed by 1.5B and 7B but solved by 32B model
- **Level 4 (Hardest)**: Problems failed by all three models

### E.5 DATA FORMAT CONVERSION

After categorization, we convert the data into Hugging Face Dataset format for efficient training:

```python
from datasets import Dataset
import jsonlines

def convert_to_hf_format(data_path, target_path):
    data = list(jsonlines.open(data_path, mode="r"))
    dataset = Dataset.from_list(data)
    dataset.save_to_disk(target_path)
    return dataset
```

Listing 2: Dataset Format Conversion

## F TRAINING PROMPT SAMPLING STRATEGY

### F.1 GRPO CONFIGURATION

Our training employs Group Relative Policy Optimization (GRPO) with carefully designed sampling strategies. The key configuration parameters are:

Table 10: GRPO Training Configuration

| Parameter | Value |
|---|---|
| num_prompts_per_step | 32 |
| num_generations_per_prompt | 16 |
| max_rollout_turns | 1 |
| normalize_rewards | True |
| use_leave_one_out_baseline | false |
| reference_policy_kl_penalty | 0.001 |
| ratio_clip_min | 0.2 |
| ratio_clip_max | 0.2 |

### F.2 PROMPT TEMPLATE DESIGN

The prompt template is designed to encourage step-by-step reasoning:

```
Solve the following math problem. Make sure to put the answer (and only
    answer) inside \boxed{}.

{problem_statement}
```

Listing 3: Mathematical Reasoning Prompt Template

## G EVALUATION SAMPLING STRATEGY

### G.1 MULTI-SHOT EVALUATION

For robust evaluation, we implement multi-shot sampling with different repetition counts based on dataset characteristics:

Table 11: Evaluation Sampling Configuration

| Dataset | Repetitions | Sampling Strategy |
|---|---|---|
| MMLU-Pro | 2 | Temperature=0.6, TOP_P=0.95 |
| GPQA | 1 | Temperature=0.6, TOP_P=0.95 |
| SuperGPQA | 1 | Temperature=0.6, TOP_P=0.95 |
| AIME24 | 10 | Temperature=0.6, TOP_P=0.95 |
| AIME25 | 10 | Temperature=0.6, TOP_P=0.95 |
| Olympiad | 4 | Temperature=0.6, TOP_P=0.95 |
| Minerva | 4 | Temperature=0.6, TOP_P=0.95 |

### G.2   ANSWER EXTRACTION AND VERIFICATION

We implement sophisticated answer extraction mechanisms for different question types:

### G.2.1   MATHEMATICAL EXPRESSION MATCHING

For mathematical problems, we extract answers using LaTeX pattern matching:

```
1  ANSWER_PATTERN_BOXED = r"(?i)\\boxed\s*{([^\n]+)}"
2
3  def extract_mathematical_answer(response_text):
4      match = re.search(ANSWER_PATTERN_BOXED, response_text)
5      if match:
6          extracted_answer = match.group(1)
7          # Normalize the extracted answer
8          extracted_answer = normalize_response(extracted_answer)
9      return extracted_answer
```

Listing 4: Mathematical Answer Extraction

### G.2.2   MULTIPLE CHOICE ANSWER EXTRACTION

For multiple-choice questions, we extract answers using pattern matching:

```
1  ANSWER_PATTERN_MULTICHOICE = r"(?i)Answer[ \t]*:[ \t]*\$?([A-D])\$?"
2
3  def extract_multiple_choice_answer(response_text):
4      match = re.search(ANSWER_PATTERN_MULTICHOICE, response_text)
5      if match:
6          extracted_answer = match.group(1).upper()
7      return extracted_answer
```

Listing 5: Multiple Choice Answer Extraction

### G.3   EQUIVALENCE CHECKING

For problems, we implement equivalence checking using a dedicated LLM:

```
1  EQUALITY_TEMPLATE = r"""
2  Look at the following two expressions (answers to a math problem) and
       judge whether they are equivalent. Only perform trivial
       simplifications.
3
4  Examples:
5      Expression 1: $2x+3$
6      Expression 2: $3+2x$
7  Yes
8
9      Expression 1: 3/2
10     Expression 2: 1.5
```

```
11 Yes
12
13     Expression 1: $x^2+2x+1$
14     Expression 2: $(x+1)^2$
15 Yes
16
17 YOUR TASK:
18 Respond with only "Yes" or "No" (without quotes).
19
20     Expression 1: %(expression1)s
21     Expression 2: %(expression2)s
22 """
23
24 def check_equality(sampler, expr1, expr2):
25     prompt = EQUALITY_TEMPLATE % {"expression1": expr1, "expression2":
       expr2}
26     response = sampler([dict(content=prompt, role="user")])
27     return response.lower().strip() == "yes"
```

Listing 6: Equivalence Verification

## G.4 EVALUATION PIPELINE

The complete evaluation pipeline processes datasets in parallel:

---
**Algorithm 2** Evaluation Pipeline

---
**Require:** Dataset $D$, Model $M$, Evaluation Config $C$
 1: Load dataset $D$ from remote URL
 2: Shuffle examples with fixed seed for reproducibility
 3: Initialize sampler with model $M$ and config $C$
 4: **for** each example $e \in D$ **do**
 5:     Generate response $r$ using sampler
 6:     Extract answer $a$ from response $r$
 7:     Compute score $s$ based on ground truth
 8:     Store result $(e, r, a, s)$
 9: **end for**

---

# H IMPLEMENTATION DETAILS

## H.1 DETAILS IN MOTIVATION

In section 4, for each problem, we first sample $K=64$ full chain-of-thought (CoT) solutions with nucleus sampling (temperature = 0.6, top-p = 0.95, max new tokens = $32k$). Each solution is labeled as correct or incorrect by exact matching the final answer against the ground truth.

## H.2 MULTI-PROCESSING FOR DATA GENERATION

For efficient data generation, we implement multi-processing with dynamic GPU allocation:

```python
def worker_process_dynamic(proc_id, task_queue, progress_queue, config):
    # Dynamic GPU allocation based on process ID
    global_cuda_visible = os.environ.get("CUDA_VISIBLE_DEVICES", None)
    available_gpus = [x.strip() for x in global_cuda_visible.split(",")]
    assigned_gpus = []
    for i in range(config["tensor_parallel"]):
        assigned_index = (proc_id * config["tensor_parallel"] + i) % len(
    available_gpus)
        assigned_gpus.append(available_gpus[assigned_index])
    os.environ["CUDA_VISIBLE_DEVICES"] = ",".join(assigned_gpus)

    # Initialize model with assigned GPUs
    model = LLM(
        model=config["base_model_name"],
        tensor_parallel_size=config["tensor_parallel"],
        max_model_len=config["max_tokens"],
        gpu_memory_utilization=0.95,
    )
```

Listing 7: Multi-Processing Data Generation

## H.3 HYPERPARAMETER CONFIGURATION

Key hyperparameters for training:

Table 12: Training Hyperparameters

| Parameter | Value |
|---|---|
| Learning Rate | 1.0e-6 |
| Weight Decay | 0.01 |
| Batch Size | 512 |
| Max Sequence Length | 512 |
| Precision | bfloat16 |
| Temperature | 1.0 |
| Top-p | 1.0 |

# I TRAINING DYNAMIC DISCUSSION

## I.1 REASONING LENGTH REDUCTION

To make the efficiency gains concrete, we report a representative item solved by **Qwen3-8B**. Before RLPIR training, the model produced a *2802*-token solution. After RLPIR with a 512-token prefix budget, the same item was solved in 1299 tokens—**a 53.6% reduction**—while yielding the same final answer. This case study aligns with the aggregate trend reported in Section 7.3, indicating that optimizing the early prefix encourages earlier commitment to high-consistency trajectories and prunes meandering continuations.

**Qwen3-8B's answer before training: 2802tokens**

```
The question is: ## Task 2\n\nAs the special train stops, a freight
↪   train passes by. Peter counts the cars: Right after the
↪   locomotive, 5 closed cars pass. Then follow 4 times as many
↪   open cars. At the end, 6 tank cars pass.\n\nHow many cars does
↪   the freight train have?

Response:
<think>
Okay, let me try to figure out how many cars are in the freight
↪   train. So, Peter is counting the cars as the freight train
↪   passes by. Let me break down the information step by step.
[... (reasoning omitted for brevity) ...]
</think>
The freight train's composition is described in a sequence by Peter
↪   as it passes by. He counts the cars in the following order:
- **After the locomotive**, 5 **closed cars** pass.
- Then, **4 times as many open cars** follow the closed cars.
- Finally, 6 **tank cars** conclude the train.
To determine the total number of **cars**, we first calculate the
↪   number of each type:
- Closed cars: 5
- Open cars: 4 × 5 = 20
- Tank cars: 6
Adding these together gives:
$$
5 + 20 + 6 = 31
$$
The problem specifies that Peter counts the **cars**, and since the
↪   locomotive is mentioned as a separate entity ("right after the
↪   locomotive"), it is **not** included in the count of cars.
---
**Final Answer:**
$$
\boxed{31}
$$
```

**Qwen3-8B's answer after RLPIR training: 1299 tokens**

```
<think>
Okay, let's see. So the problem is about Peter counting the cars of
↪   a freight train as it passes by. The question is asking how
↪   many cars the freight train has in total. Let me break down the
↪   information given step by step.
[... (reasoning omitted for brevity) ...]
</think>
To determine the total number of cars in the freight train, we
↪   analyze the sequence as described:
1. **After the locomotive**, there are **5 closed cars**.
2. **Then**, there are **4 times as many open cars** as closed
↪   cars. Since there are 5 closed cars, this means:
   $ 4 \times 5 = 20 $ open cars.
3. **At the end**, there are **6 tank cars**.
Adding these together gives the total number of cars:
$$
5 \text{ (closed)} + 20 \text{ (open)} + 6 \text{ (tank)} = 31
$$
Thus, the freight train has:
$$
\boxed{31}
$$
```

---

**Qwen3-8B's answer after GRPO training: 3271**

```
The question is: ## Task 2\n\nAs the special train stops, a freight
↪  train passes by. Peter counts the cars: Right after the
↪  locomotive, 5 closed cars pass. Then follow 4 times as many
↪  open cars. At the end, 6 tank cars pass.\n\nHow many cars does
↪  the freight train have?

Response:
<think>
Okay, let's see. So the problem is about Peter counting the cars of
↪  a freight train as it passes by. The question is asking how
↪  many cars the freight train has in total. Let me try to break
↪  down the information given
step by step.
[... (reasoning omitted for brevity) ...]
</think>.
To determine how many **cars** the freight train has, we need to
↪  carefully analyze the sequence of events described:
---
### Step-by-Step Breakdown
1. **After the locomotive**, Peter observes **5 closed cars**.
   - These are the first set of cars.
   - So, we count **5** cars.
2. **Then follow 4 times as many open cars**.
   - This means the number of open cars is **4 times** the number
   ↪  of closed cars.
   - Since there are 5 closed cars, the number of open cars is:
     $$
     4 \times 5 = 20
     $$
   - So, we add **20** more cars.
3. **At the end**, **6 tank cars** pass by.
   - This is the final set of cars.
   - Add **6** more cars.
---
### Total Number of Cars
Now, summing up all the cars (excluding the locomotive, as the
↪  problem refers to "cars" and the locomotive is a separate
↪  unit):
$$
5 \text{ (closed cars)} + 20 \text{ (open cars)} + 6 \text{ (tank
↪  cars)} = 31
$$
---
### Final Answer
$$
\boxed{31}
$$
```

## I.2 SIMILARITY IMPROVEMENT

To verify that RLPIR effectively increases intra-group semantic agreement, we track the average pairwise cosine similarity among sampled prefixes during training. We observe a steady upward trend in similarity as optimization progresses, indicating that the policy learns to generate more coherent and self-consistent reasoning prefixes over time. This improvement confirms that the intra-group consistency reward provides a strong learning signal and successfully guides the model toward producing reasoning trajectories that are both semantically aligned and mutually consistent.

## I.3 COLLAPSE PHENOMENON WITHOUT ASYMMETRIC ADVANTAGES

We further analyze the behavior of RLPIR when the asymmetric advantage clipping in Eq. 8 is removed. In this setting, the model rapidly drives similarity to an extreme by generating nearly

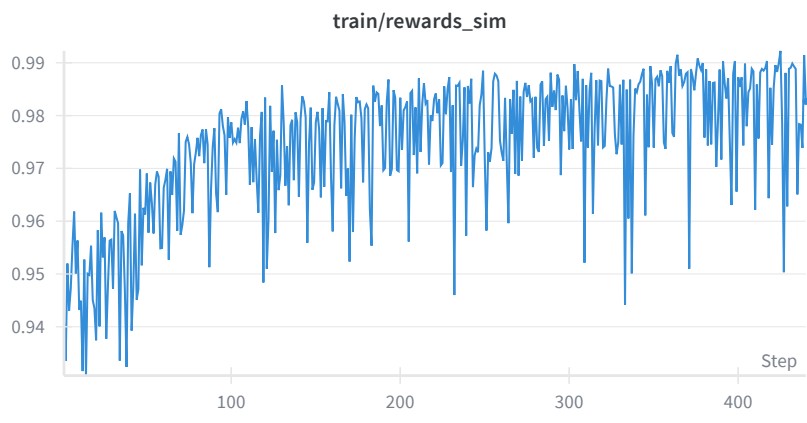

Figure 4: Average pairwise cosine similarity among sampled reasoning prefixes during training. The clear upward trend indicates that RLPIR enhances intra-group semantic agreement and encourages the generation of more coherent and self-consistent reasoning trajectories.

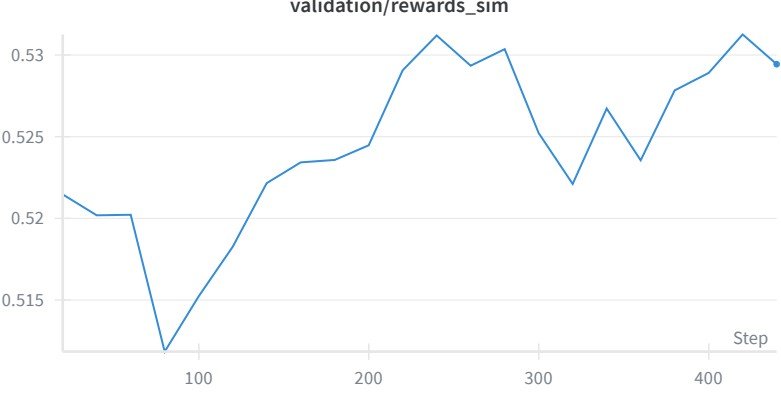

Figure 5: Trend of average pairwise cosine similarity on the AIME24 benchmark. The consistent increase demonstrates that the model progressively learns to produce reasoning prefixes that are semantically closer and more coherent as training advances.

identical and repetitive continuations, leading to a collapse of diversity. This repetition causes a sharp drop in task accuracy and results in drastically shortened outputs. Such "mode collapse" behavior highlights the importance of penalizing only low-consistency samples: the Asymmetric design regularizes the policy, maintains useful variability in reasoning trajectories, and prevents degenerate solutions that maximize similarity without improving actual reasoning quality.

---

**Collapse Phenomenon without One-Sided Advantages**

```
The question is: ## Task 2\n\nAs the special train stops, a freight
↪   train passes by. Peter counts the cars: Right after the
↪   locomotive, 5 closed cars pass. Then follow 4 times as many
↪   open cars. At the end, 6 tank cars pass.\n\nHow many cars does
↪   the freight train have?

Response:
<think>
Okay, let's see. Let me try to break down the information given
↪   step by step. Let me try to break down the information given
↪   step by step. Let me try to break down the information given
↪   step by step. Let me try to break down the information given
↪   step by step. Let me try to break down the information given
↪   step by step. Let me try to break down the information given
↪   step by step. Let me try to break down the information given
↪   step by step. Let me try to break down the information given
↪   step by step. Let me try to break down the information given
↪   step by step. Let me try to break down the information given
↪   step by step. Let me try to break down the information given
↪   step by step. Let me try to break down the information given
↪   step by step. Let me try to break down the information given
↪   step by step. Let me try to break down the information given
↪   step by step. [... (repeat reasoning content omitted for
↪   brevity) ...]
```

---

## J  MORE EXPERIMENTS

### J.1  EFFECT OF EMBEDDING METHOD

To assess the sensitivity of our intrinsic reward to the embedding model, we ablate three embedding method. As shown in Table 13, the performance of **RLPIR** is remarkably robust to the choice of embedding.

| Embedding Method | Acc↑ | Len(tokens) |
|---|---|---|
| all-MiniLM-L6-v2 | **78.8** | 11797 |
| Qwen3-Embedding-0.6B | 78.6 | **11239** |
| TF-IDF | 78.1 | 12016 |

Table 13: Ablation study on the embedding method for computing intra-group consistency. Results are on AIME24 with the Qwen3-8B model. Performance is stable across different embedding methods.

