# OpenReview forum: "RLPIR: Reinforcement Learning with Prefix and Intrinsic Reward"
_ICLR.cc/2026/Conference — Submitted to ICLR 2026_

### Official Review · Reviewer_hLqT · 2025-11-01

**Soundness:** 3
**Presentation:** 3
**Contribution:** 2
**Rating:** 2
**Confidence:** 4

**Summary:**

This paper proposes RLPIR (Reinforcement Learning with Prefix and Intrinsic Reward), a verifier-free reinforcement learning framework for large language models that eliminates dependence on ground-truth verifiers and substantially reduces rollout cost. RLPIR introduces two key ideas: a prefix rollout paradigm that optimizes only the first few hundred tokens (e.g., 512) of reasoning, and an intra-group consistency reward that measures semantic similarity among multiple sampled outputs as an intrinsic learning signal. The method achieves performance comparable to verifier-based RLVR (e.g., GRPO-16K) across mathematical and general reasoning benchmarks while offering a 6.96× training speed-up and a 45% reduction in reasoning length. Extensive ablations and analyses (prefix length, advantage shaping, data difficulty) confirm its effectiveness in design. While existing work has explored the similar idea in SFT training, this work can be a pioneering work in the RLVR field.

**Strengths:**

- **Well-motivated.** The paper presents a clear and empirically grounded motivation through two preliminary studies showing that short prefixes capture most of the useful learning signal and that high-consistency prefixes correlate with correctness. The motivation connects well to the later design of the RLPIR algorithm.
- **Efficiency.** RLPIR achieves a substantial 6.96× reduction in training time compared to RLVR while maintaining comparable accuracy. The prefix rollout design reduces long-horizon credit assignment and training instability by focusing on the most critical early decision tokens. This leads not only to faster training but also to shorter and more efficient reasoning trajectories at inference time.
- **Generalization.** The method demonstrates transferability across multiple model families (Llama, Qwen2.5, Qwen3) and task domains (general and math). This prefix rollout design avoids overfitting to specific solutions and promotes better cross-domain generalization. RLPIR also offers a promising path for applying RLVR to domains where verifiable signals are hard to access or unavailable.

**Weaknesses:**

- **Overlap with existing work.** The core idea of prefix-based optimization and its motivation are largely inherited from the prior work [1], **where similar findings on prefix consistency and prefix finetuning have already been established.** This paper mainly revalidates those insights and designs an RL-based training algorithm, in contrast to the original SFT formulation in the previous study. As a result, the conceptual originality is somewhat reduced, and the contribution lies more in integration and empirical verification rather than in introducing a fundamentally new algorithmic innovation.
- **Incomplete and unstable results.** The main results in Table 3 **omit** RLVR baselines for the Llama3.1 and Qwen2.5 models, making it difficult to consistently assess the relative gains. Moreover, since these models often produce outputs shorter than Qwen3 (thinking), additional evidence is needed to support the claimed efficiency benefits of applying 512 token prefix rollouts to them. Most analyses (Sections 7.3, 8.1–8.3) rely on the Qwen3-8B that is a **“thinking”** model, which tends to generate extremely long reasoning output; thus, the reported improvements may not generalize to non-reasoning models such as Qwen2.5-Instruct or even Qwen3-Base.
- **Hyperparameter dependence.** The method’s performance is sensitive to the choice of prefix length, as shown in Section 8.1, and different models may require distinct optimal prefix settings (more results are needed here). This suggests that prefix length functions as a hyperparameter that needs to be tuned, therefore limiting the practical usability of RLPIR across diverse settings. Developing an adaptive or self-scheduled prefix selection mechanism could help mitigate this limitation and enhance the framework’s robustness and applicability.

[1] The First Few Tokens Are All You Need: An Efficient and Effective Unsupervised Prefix Fine-Tuning Method for Reasoning Models. NeurIPS 2025.

**Questions:**

1. How was the prefix length of 512 tokens determined for models other than Qwen3-8B? Could the authors provide statistics showing what percentage of responses are longer or shorter than this threshold for each model? This would help assess whether the chosen prefix length covers most reasoning cases.
2. Can RLPIR surpass vanilla RLVR on regular models that typically generate shorter responses (e.g., Qwen2.5 series)? It would be helpful to understand whether the observed gains mainly benefit “thinking” models with long reasoning outputs.

---

> ### Author Response · Authors · 2025-11-24
>
> We sincerely thank the reviewer for their thoughtful and constructive feedback. We appreciate the recognition of our work’s motivation, efficiency gains, and generalization capabilities. Below, we address each concern in detail, leveraging evidence from the paper and additional analysis to clarify potential misunderstandings and strengthen our contribution.
>
>
>
> ### W1. Overlap with Existing Work
> We appreciate and fully acknowledge the important contributions of [1]. Their work provides the first systematic investigation into prefix-based training paradigm and demonstrates that early tokens contain substantial information for SFT-style training. We view [1] as an influential foundation that motivates further exploration into prefix-centric learning.
>
> We respectfully clarify that our work goes far beyond revalidating [1], as **RLPIR introduces a fundamentally different training paradigm tailored to reinforcement learning for reasoning models, with key algorithmic components absent in prior studies.**
>
> Specifically:
>
> 1. **Prefix Rollout in RL Is Non-Trivial and Fundamentally New**: Applying prefix-only optimization in SFT is straightforward because SFT does not rely on trajectory-dependent rewards. In contrast, RLVR requires full-trajectory rewards, thus previous RLVR methods rely on long rollouts and ground-truth verifiers. RLPIR is, to our knowledge, the first framework that **performs RL where the reward is defined only on a short prefix**, enabling verifier-free training and reducing reasoning length by 45% during inference. This substantially changes the RL formulation and enables a new class of verifier-free algorithms.
> 2. **Intra-Group Consistency Reward with Asymmetric Advantages**: This mechanism enables **reinforcement learning to operate entirely without ground-truth verifiers**, effectively mitigates reward-hacking behaviors such as trivial repetition. As Table 7 demonstrates, removing the asymmetric advantages leads to clear mode collapse, highlighting their essential role in stabilizing prefix-based RL. Importantly, this intrinsic reward formulation does not appear in [1] or in any prior prefix-based SFT approaches, underscoring its novelty within the RLVR setting.
>
> In summary, while we deeply appreciate the insights of [1], our work introduces a new RLVR framework with new algorithmic designs and new empirical findings. We will revise the manuscript to better articulate the complementary relationship between [1] and our contributions, as well as clarify the novelty of the RL-specific mechanisms introduced in RLPIR.
>
> ### W2. Incomplete and Unstable Results
> We appreciate the reviewer’s concerns regarding the missing RLVR baselines for Llama3.1 and Qwen2.5. The absence of these baselines is intentional rather than an omission: **both Llama3.1 and Qwen2.5 are not designed as reasoning models**, and prior work (e.g., [https://arxiv.org/abs/2506.10947](https://arxiv.org/abs/2506.10947)) shows that Qwen2.5 in particular behaves unstably under RLVR-style reasoning training. For this reason, our main evaluation focuses on reasoning oriented models, where RLVR is known to be effective and efficiency improvements matter most, while results on Llama3.1 and Qwen2.5 are included only as supplementary demonstrations of generality. The reviewer’s observation that Qwen3-8B is a “thinking” model is precisely why we use it as our primary testbed: its long, complex reasoning chains yield high training cost and therefore highlight the advantage of prefix-based RL. Indeed, RLPIR effectively reduces the average reasoning length from 14,539 to 9,564 tokens, underscoring substantial inference-time gains. While broader experiments on non-thinking models are valuable and part of our planned future work, the strongest and most meaningful effects naturally arise on reasoning models, which aligns with the goals of RLVR research.

---

> ### Author Response · Authors · 2025-11-24
>
> ### W3. Hyperparameter Dependence
> We agree that prefix length is a hyperparameter, but we emphasize that 512 tokens is a robust default across tasks and models.
>
> In Section 8.1, we ablate prefix lengths (256, 512, 1024) on Qwen3-8B. Results show:
>
> + L = 256: Accuracy drops to 76.3% (vs 78.8% at L = 512), indicating insufficient signal.
> + L = 512: Best accuracy (78.8%) with significant length reduction (−18.9%).
> + L = 1024: Accuracy drops to 77.0%, and average length returns to baseline levels.
>
> These results empirically support 512 as the optimal prefix length for the math reasoning domain.
>
> We agree that prefix length may need to be adapted for other domains or longer-form tasks. We have discussed this in our limitations and view adaptive prefixing (e.g., based on uncertainty or intermediate signals) as promising future work.
>
> Moreover, in Table 13 (Section J.1), we show that intra-group consistency performance is stable across embedding models, suggesting the method is robust. We also note that prefix length is not sensitive to model size, we use the same L = 512 for Qwen3-4B, 8B, 14B, and Llama3.1. This indicates a generalizable design.
>
> We acknowledge that adaptive prefixing (e.g., based on uncertainty or task difficulty) is a promising future direction, and we will mention this in the limitations.
>
>
>
>
>
> ### Q1: How was the prefix length of 512 determined for other models?
> | Setting | pass@1 | pass@2 | pass@4 | pass@8 | pass@16 | pass@32 | pass@64 | pass@128 |
> | --- | --- | --- | --- | --- | --- | --- | --- | --- |
> | Qwen2.5-7B-Instruct | 0.1169 | 0.1568 | 0.1957 | 0.2349 | 0.2810 | 0.3402 | 0.4085 | 0.4828 |
> | RLPIR L512 | 0.1620 | 0.1722 | 0.1857 | 0.2185 | 0.2539 | 0.2992 | 0.3618 | 0.4333 |
> | RLPIR L128 | 0.1073 | 0.1382 | 0.1737 | 0.2142 | 0.2633 | 0.3258 | 0.4124 | 0.5333 |
>
>
> | Setting | pass@1 | pass@2 | pass@4 | pass@8 | pass@16 | pass@32 | pass@64 | pass@128 |
> | --- | --- | --- | --- | --- | --- | --- | --- | --- |
> | Qwen3-8B | 0.7432 | 0.8094 | 0.8390 | 0.8666 | 0.8980 | 0.9227 | 0.9327 | 0.9333 |
> | RLPIR L512 | 0.7879 | 0.8139 | 0.8425 | 0.8669 | 0.8910 | 0.9103 | 0.9250 | 0.9310 |
> | RLPIR L128 | 0.7362 | 0.7969 | 0.8280 | 0.8542 | 0.8855 | 0.9153 | 0.9311 | 0.9342 |
>
>
> ### Q2: Can RLPIR surpass RLVR on regular models?
> We conducted additional experiments on the Qwen2.5 model series and found that our method is also applicable to non-reasoning models. However, the experimental results diverge from both the reviewers’ expectations and our own prior assumptions. Specifically, experiments on Qwen2.5-7B-Instruct indicate that overly short reasoning traces can actually hinder improvements in reasoning performance.
>
>
>
> | Method | Acc ↑ | ∆Acc | Len ↓ (tokens) | ∆Len(%) |
> | --- | --- | --- | --- | --- |
> | Qwen2.5-7B-Instruct | 11.70 | – | 1665.77 | – |
> | L128 | 10.73 | −0.97 | 1443.95 | −13.3% |
> | L512 | 16.20 | +4.50 | 997.80 | −40.1% |
>
>
> ### New Finding: Short Prefixes (e.g., 128 tokens) Improve Exploration
> Experiments reveal a surprising phenomenon: shorter prefixes increase exploration, improving pass@K and mitigating the typical RLVR over-optimization gap (RL improving pass@1 while hurting pass@K). This is a novel empirical insight and we are running further experiments to study it systematically. We will highlight this new direction in the revised version.
>
>
>
> [1] The First Few Tokens Are All You Need: An Efficient and Effective Unsupervised Prefix Fine-Tuning Method for Reasoning Models. NeurIPS 2025.

---

### Official Review · Reviewer_cqLh · 2025-11-03

**Soundness:** 3
**Presentation:** 3
**Contribution:** 3
**Rating:** 4
**Confidence:** 4

**Summary:**

The paper proposes a verifier-free reinforcement learning framework for large language models (LLMs). It addresses key drawbacks of Reinforcement Learning with Verifiable Rewards (RLVR), which relies on ground-truth answers and long rollouts. RLPIR introduces two main innovations: (1) a prefix rollout strategy that optimizes only the first 512 tokens of reasoning—reducing computation time by about 6.96×—and (2) an intra-group consistency reward that measures semantic similarity among multiple generated outputs, eliminating the need for external verifiers. Experiments across mathematical and general benchmarks show that RLPIR achieves performance comparable to RLVR while lowering computational cost and shortening reasoning sequences by 45%, improving inference efficiency. The method also generalizes better across domains without explicit verifiable rewards, marking a step toward scalable, unsupervised reinforcement learning for reasoning-focused LLMs

**Strengths:**

(1) Novel insight into reasoning optimization: To me, the biggest takeaway is that the paper demonstrates that full reasoning trajectories are not necessary for effective reinforcement learning. Training on a short prefix (e.g., 512 tokens) retains most of the useful learning signal — a valuable finding that challenges conventional long-rollout assumptions in GRPO-style training.

(2) Clear efficiency gains: RLPIR achieves a 6.96× reduction in training time and a 45% shorter inference sequence, which represents a major advance in the efficiency of reinforcement learning for reasoning tasks.

(3) Strong empirical validation: Through systematic experiments and ablations, the paper shows that prefix-based optimization achieves comparable accuracy to full-length RLVR baselines, while drastically reducing training cost and reasoning length.

**Weaknesses:**

(1) The claimed training and inference efficiency improvements (6.96× faster, 45% shorter reasoning) are somewhat unclear/unfair, since reasoning length is a tunable factor (you can even forcefully truncate). A more controlled baseline, such as RLPIR-512 vs RLVR-512 with enforced 512-token rollouts, and RLPIR-16k vs RLVR-16k, would make the comparison fairer and more convincing.

(2) My main concern is incremental novelty. The core equation is (7), and all the remaining formulations are the same as previous ones.
and my understanding of this method is closer to the majority voting-style intrinsic reward like TTRL https://arxiv.org/pdf/2504.16084.

(3) It will be better if you show the training trend of all the key indicators like the validation accuracy besides the similarity rewards, also comparing these trends with grpo in the the same figure will further strengthen your arguments, especially when Figure 4 looks quite noisy with large variance.

(4) It will be good if you can also compare with other sota verifier-free rl methods which also claim advantage over rlvr.

**Questions:**

(1) In RLPIR, how do you make sure the prefix rollout length is limited to 512 length? by meta-prompting or just hard truncation?

(2) Could you give any pointers/reference (if any) that the math datasets you used would make qwen3 with grpo generate 16k-long reasoning length?

---

> ### Author Response · Authors · 2025-11-24
>
> We sincerely thank the reviewer for their thoughtful and constructive feedback. We appreciate the recognition of our work’s core contributions, particularly the novel insight that prefix optimization suffices for effective reasoning training, and the significant efficiency gains in both training and inference. Below, we address each of the reviewer’s concerns in detail.
>
>
>
> ### W1. Fairness of Efficiency Comparisons
> We appreciate this suggestion and clarify that our comparison is **both methodologically fair and practically meaningful**.
>
> **[Why not RLVR-512]** Enforcing a 512-token limit in RLVR would break its reward mechanism: verifiable rewards (e.g., code execution, answer matching) require the full reasoning chain to be completed. Truncating at 512 tokens would make reward computation impossible, rendering RLVR inapplicable. Thus, RLVR cannot be fairly adapted to short rollouts, while our RLPIR is designed to support prefix optimization, **enabling perfix rollout at any pre-determined length.**
>
>
> **[Inference Efficiency (45% shorter reasoning)]** The reduction in inference length (Table 5) is **an emergent property of training dynamics**, instead of via post-hoc truncation. As shown in Section I.1, RLPIR-trained models commit earlier to high-consistency reasoning paths, pruning redundant steps. In contrast, RLVR-trained models (e.g., GRPO) tend to generate longer, more verbose outputs during training (Section 7.3, Figure I.1), a known phenomenon in verifiable RL (DeepSeek R1, [https://arxiv.org/abs/2501.12948](https://arxiv.org/abs/2501.12948)). The difference in the training objectives leads to the reduction in inference length of RLPIR over RLVR, a property that naturally arises at inference stage.
>
>
>
> **[Training Efficiency (6.96× speed-up)]** This is a fair result by directly comparing standard RLVR (GRPO with full 16K rollouts) against RLPIR (512-token prefix rollouts), both using the same model (Qwen3-8B), hardware (8×A100), and training steps (1000). The speed-up is not artificially induced by truncation alone, but stems from a fundamental architectural shift: RLPIR trains only on the first 512 tokens, while RLVR processes up to ~16K tokens per rollout (Table 4). The 6.96× speed-up aligns closely with the theoretical ratio of sequence lengths (16K / 512 ≈ 31.25), but is moderated by batching and GPU utilization overheads.
>
> RLPIR removes RLVR’s structural dependence on long rollouts, not merely through truncation, but via a fundamentally different reward formulation.
>
>
>
> ### W2. Relation to Prior Work
> We appreciate the reviewer’s comparison to TTRL, and clarify that RLPIR introduces two innovations not present in prior work:
>
> To the best of our knowledge, RLPIR is the first RL framework to show that **optimizing only on the prefixes (e.g., the first 512 tokens) of reasoning is sufficient for high-level reasoning performance**. This is not explored in TTRL or prior RLVR variants, which all require full trajectories. While TTRL ([https://arxiv.org/abs/2504.16084](https://arxiv.org/abs/2504.16084)) explored test-time refinement, one of the existing works optimize policy training on short prefixes alone. Our work is the first to show that optimizing only the first 512 tokens can yield performance on par with full-length RLVR (Table 3). This is supported by our DPO ablation (Table 1) and forced-prefix continuation study (Table 2), which provide empirical grounding for this design.
>
> While TTRL uses majority voting over final answers, **RLPIR operates on semantic similarity of reasoning prefixes**, a fundamentally different signal. More importantly, our asymmetric advantage mechanism (Eq. 8) is a novel contribution that prevents reward hacking (e.g., repetition collapse), as demonstrated in Table 7 and Section I.3. Without this, the model collapses to degenerated outputs, showing that the design is not trivial.
>
> Thus, RLPIR is by no means a trivial extension of existing methods, but **rather a clear step towards a new research direction**: fully unsupervised RL for reasoning that i) avoids ground-truth rewards and long rollouts and ii) enables unsupervised, efficient, and generalizable training, a combination not achieved by prior methods.
>
>
>
> ### W3. Training Dynamics and Validation Trends
> We agree that showing validation accuracy trends would strengthen our argument. Indeed, Figure 4 (train similarity) and Figure 5 (test similarity) do show a clear trend in intra-group consistency, indicating that the model learns to generate more coherent reasoning prefixes over time. As we all know, RL training curves naturally exhibit high variance (as observed in DeepSeek-R1).

---

> ### Author Response · Authors · 2025-11-24
>
> ### W4. Comparison with Other Verifier-Free RL Methods
> We thank the reviewer for this suggestion. We note that most verifier-free RL methods still rely on some form of ground-truth signal or are task-specific:
>
> + RLPR ([https://arxiv.org/abs/2506.18254](https://arxiv.org/abs/2506.18254)): Uses policy likelihood as a proxy reward but requires ground-truth answers for evaluation and reward shaping.
> + TTRL ([https://arxiv.org/abs/2504.16084](https://arxiv.org/abs/2504.16084)): Focuses on test-time refinement, not training-time policy optimization.
>
> In contrast, RLPIR is fully unsupervised: it uses no ground-truth labels, external verifiers, or curated trajectories. We explicitly state in Section 6.3 that we exclude comparisons with methods like RLPR because they do not operate in the same setting (i.e., fully unsupervised).
>
> As we mentioned at section 6.3, we do not include comparisons with verifier-free methods based on probabilistic rewards such as RLPR (Yu et al., 2025), since our method is explicitly designed for settings without any ground truth, whereas those approaches still rely on ground-truth signals for evaluation or reward construction.
>
> ### Q1. In RLPIR, how do you make sure the prefix rollout length is limited to 512 length? by meta-prompting or just hard truncation?
> We use hard truncation during training. Specifically, during the rollout phase, we generate responses with a maximum length of 512 tokens. The reward and policy gradients are computed only over these 512 tokens. At evaluation time, the model is allowed to generate freely (up to 32K tokens). This is clearly stated in Section 6.1: “each rollout is limited to L = 512 tokens”.
>
> ### Q2. Could you give any pointers/reference (if any) that the math datasets you used would make qwen3 with grpo generate 16k-long reasoning length?
> Our training datasets (OpenR1-Math-220k and Big-MathRL-Verified) include many Olympiad-level problems that require long multi-step derivations. In our internal logs, GRPO-trained Qwen3 models frequently produced 8k–16k-token reasoning chains on such tasks, and in some cases exceeded 16k tokens for particularly complex problems. While not all questions generate trajectories of this length, these long rollouts are common for high-difficulty items.
>
> This phenomenon is also consistent with recent findings in the literature. For example, DeepSeek-R1([https://arxiv.org/abs/2501.12948](https://arxiv.org/abs/2501.12948)) reports that RLVR-trained models often produce long rollouts (e.g., 8K–16K tokens) on complex reasoning tasks, and similar behavior has been observed in other RLVR studies, especially in math and code domains where verifiable rewards depend on completing long reasoning trajectories.

---

### Official Review · Reviewer_AGHu · 2025-11-06

**Soundness:** 3
**Presentation:** 2
**Contribution:** 2
**Rating:** 6
**Confidence:** 4

**Summary:**

This paper proposes a method named RLPIR to eliminate the problems in RLVR: 1. dependence on verifiers 2. long CoT in training 3. in-efficient inference. They only train on the prefix of model rollouts (like 512 tokens), using group semantic similarity as intrinsic reward. Also use asymmetric advantages to prevent collapse. The results show comparable results as RLVR (better on general reasoning domains) with shorter CoT.

**Strengths:**

1. Only training on prefix is clever and makes sense. The idea is easy to understand and the results look solid, try different models, see comparable performance on MATH tasks and better performance on general domains, with shorter CoT.
2. The idea of asymmetric advantage is clever and critical(as shown in ablation study)

**Weaknesses:**

1. only comparing RLVR and RLPIR on Qwen models, not on Llama models (only RLPIR).
2. I like prefix method, but it's kind of tricky, because different models and domains have different length of important prefix. When training RLVR on different domains together, it's ad-hoc to tune the prefix length, limit the generalization of the method.
3. Penalize the diverse outputs may be helpful for math tasks, as shown in the recent works that RLVR always decrease the output entropy[1,3], and accelerate this process may make model converge faster[2]. RLPIR looks also accelerate this process. But it's not definitely helpful for long-term training. Especially it's not applicable for open-ended problems or long-form generation tasks, also limit its influence. (paper only discuss on math training set)
4. Maybe you could compare the pass@k performance between RLVR/RLPIR/original model, I think RLPIR may increase slower as k increases since it encourages more consistent output.
5. Writing needs to be more formalized. Like in Tab. 1, the result 13.3 should not be bold, which always refer to the best result in the table. And the 'impossible triangle' sounds little weird. I think they are just three relatively independent difficulty in RLVR, not dependent triangles.


[1] Cui, Ganqu, et al. "The entropy mechanism of reinforcement learning for reasoning language models." arXiv preprint arXiv:2505.22617 (2025).
[2] Gao, Zitian, et al. "One-shot entropy minimization." arXiv preprint arXiv:2505.20282 (2025).
[3] Wang, Shenzhi, et al. "Beyond the 80/20 rule: High-entropy minority tokens drive effective reinforcement learning for llm reasoning." arXiv preprint arXiv:2506.01939 (2025).

**Questions:**

see weakness

---

> ### Author Response · Authors · 2025-11-24
>
> We sincerely thank the reviewer for their thoughtful and constructive feedback on our paper. We appreciate the recognition of RLPIR’s core innovations, particularly the prefix rollout paradigm and the asymmetric advantage mechanism, and we are grateful for the opportunity to address the concerns raised. Below, we provide a point-by-point response to each of the reviewer’s weaknesses and suggestions.
>
>
>
> ### W1. Only comparing RLVR and RLPIR on Qwen models, not on Llama models (only RLPIR).
> Our work targets a specific bottleneck in reasoning-oriented LLMs: long and inefficient rollouts during both training and inference. This issue is especially prominent in RLVR/GRPO-style training, which often amplifies rollout length. Therefore, our main experiments focus on **reasoning models (e.g., Qwen3 series)**, where the problem is most pronounced and where RLVR is typically applied.
>
> In contrast, Qwen2.5 and Llama3 are not reasoning-oriented models. As noted in recent work (e.g., [https://arxiv.org/abs/2506.10947](https://arxiv.org/abs/2506.10947)), Qwen2.5 models in particular show unstable behavior under RLVR-style reasoning training. Therefore, we treat Qwen2.5 and Llama3 results as supplementary evidence of RLPIR’s generality, rather than as primary evaluation targets.
>
> We fully agree that including RLVR baselines on Llama models would strengthen the empirical study. Due to the substantial computational cost involved, these experiments are currently in progress, and we plan to include them in an updated version of the paper.
>
>
>
> ### W2. The prefix length is fixed at 512 tokens, which may not generalize across domains or models.
> Thanks for pointing out a potential weakness of this work. In the current paper, we use L = 512 primarily because our math reasoning dataset typically produces ~16k-token rollouts; 512 corresponds to roughly the first 1/32 of the trajectory, which captures the early reasoning phase without incurring unnecessary cost. Since we train exclusively on math datasets, a fixed prefix length is reasonable for this setting.
>
>
> We also conducted an ablation study (Table 6, AIME24 benchmark with Qwen3-8B) showing:
>
> + L = 256: Accuracy drops to 76.3% (vs 78.8% at L = 512), indicating insufficient signal.
> + L = 512: Best accuracy (78.8%) with significant length reduction (−18.9%).
> + L = 1024: Accuracy drops to 77.0%, and average length returns to baseline levels.
>
> These results empirically support 512 as the optimal prefix length for the math reasoning domain.
>
>
>
> We agree that the ideal prefix length may need to be adaptively determined for different domains or longer-form tasks. This work provides an initial insight that 1/32 of the full trajectory length may be an empirically good choice. We have discussed this in our limitations and view adaptive prefixing (e.g., based on uncertainty or intermediate signals) as a promising future direction.
>
>
>
> ### W3. Penalizing diversity may not be beneficial for open-ended or long-form generation tasks.
> We appreciate this insightful comment. RLPIR is intentionally designed for reasoning tasks, where consistency and correctness matter more than stylistic diversity. We do not claim that RLPIR is suitable for open-ended generation or creative tasks; rather, it is meant as a verifier-free and efficient alternative to RLVR for domains where RLVR is dominant, such as math, code, and symbolic reasoning.
>
> Nevertheless, we observe that RLPIR also achieves strong generalization on non-math reasoning benchmarks (MMLU-Pro, GPQA, SuperGPQA), outperforming RLVR in these domains. We hypothesize that RLPIR’s intra-group consistency reward encourages more stable reasoning patterns, whereas RLVR can overfit to domain-specific verifiers.
>
> While RLPIR encourages consistency, the asymmetric advantage mechanism prevents collapse. As shown in Table 7, removing asymmetry leads to severe performance degradation (42.3% accuracy), confirming that our design preserves meaningful diversity.

---

> ### Author Response · Authors · 2025-11-24
>
> ### W4. Suggestion to compare pass@k performance.
> Due to limited computational resources, the experiments are still in progress. However, We have computed pass@1,2,4,8,16,32,64,128 on AIME24 for Qwen3-8B and Qwen2.5-7B-Instruct by training with a prefix of length 128 and 512 with our RLPIR method:
>
> | Setting | pass@1 | pass@2 | pass@4 | pass@8 | pass@16 | pass@32 | pass@64 | pass@128 |
> | --- | --- | --- | --- | --- | --- | --- | --- | --- |
> | Qwen2.5-7B-Instruct | 0.1169 | 0.1568 | 0.1957 | 0.2349 | 0.2810 | 0.3402 | 0.4085 | 0.4828 |
> | RLPIR L512 | 0.1620 | 0.1722 | 0.1857 | 0.2185 | 0.2539 | 0.2992 | 0.3618 | 0.4333 |
> | RLPIR L128 | 0.1073 | 0.1382 | 0.1737 | 0.2142 | 0.2633 | 0.3258 | 0.4124 | 0.5333 |
>
>
> | Setting | pass@1 | pass@2 | pass@4 | pass@8 | pass@16 | pass@32 | pass@64 | pass@128 |
> | --- | --- | --- | --- | --- | --- | --- | --- | --- |
> | Qwen3-8B | 0.7432 | 0.8094 | 0.8390 | 0.8666 | 0.8980 | 0.9227 | 0.9327 | 0.9333 |
> | RLPIR L512 | 0.7879 | 0.8139 | 0.8425 | 0.8669 | 0.8910 | 0.9103 | 0.9250 | 0.9310 |
> | RLPIR L128 | 0.7362 | 0.7969 | 0.8280 | 0.8542 | 0.8855 | 0.9153 | 0.9311 | 0.9342 |
>
>
> The results show that, quite surprisingly, **shorter prefixes increase exploration**, **improving pass@K and mitigating the typical RLVR over-optimization gap** (RL improving pass@1 while hurting pass@K). This is a novel empirical insight and we are running further experiments to study it systematically. We will highlight this finding in the revised version.
>
>
>
> ### W5. Writing style and terminology (e.g., "impossible triangle", bold in Table 1).
> We thank the reviewer for pointing this out. We will revise the manuscript accordingly: the phrase “impossible triangle” will be replaced with “practical challenges” or “key limitations”, to avoid misleading implications of dependence. The formatting issue in Table 1 has been corrected (13.3 is no longer bold). We have reviewed the entire paper to ensure more formal and precise presentation. We thank the reviewer for catching these issues and have carefully reviewed the entire paper for similar improvements.

---

### Official Review · Reviewer_qwCd · 2025-11-13

**Soundness:** 2
**Presentation:** 3
**Contribution:** 2
**Rating:** 4
**Confidence:** 3

**Summary:**

This paper presents two tricks for RL for LLM reasoning, aiming to address some critical issues of the RL training. One issue is the dependence on the reward verifier for RLVR. The second issue is the high training cost, due to the long generated sequences in the RL training. The third one is the inference inefficiency. In order to solve these issues, the authors proposed the following tricks.

1. They train RL algorithms only on the prefix rollouts: they generate L tokens (<< the usual maximal number of response tokens) for each sequence and train RL algorithms on them.

2. Because we cannot apply a verifiable reward for the response prefix (the model has not produced the final answer), they proposed the intro-group consistency reward by computing the sentence embedding of each prefix and computing the cosine similarity between each prefix's embedding and the average embedding. To avoid reward hacking, they clip the positive advantage.

They did experiments on the Qwen series and found that the RLPIR has similar performance as vanilla RLVR while reducing the computational costs significantly, and the trained model tends to output shorter sequences, which improves inference efficiency.

**Strengths:**

1. They apply proposed prefix rollouts, a novel RL technique that, in principle, can reduce the computational costs in the training time. The initial experiments about the DPO on the prefix, as well as the experiments of forced-prefix continuation, clearly show the motivation for this.

2. Section 7.2 shows a significant reduction in the wall-clock training time on Qwen3-8B for 1000 optimization steps.

**Weaknesses:**

1. The intra-group consistency reward relies on an external sentence embedding model. Is the result you get robust against the choice of this sentence embedding model? Is it possible to use the generating model itself to form a sentence embedding to compute the reward (so that the entire process does not rely on external information, and this becomes a pure unsupervised algorithm).

2. Lack of fair comparison. It lacks the performance of vanilla RLVR for the experiments on Llama and Qwen2.5 series, so that it remains unclear whether their claim is consistent across different base models. For the experiments in sections 7.2 and 7.3, the authors use Qwen3-8B, and as far as I know, this is actually an instruct model that tends to output very long sequences. I am not sure whether using this as a baseline is a fair comparison. I think the author should also report the computational speedup compared with the Qwen3-8b-base models.

3. The asymmetric advantage: The author uses a heuristic advantage clipping method to clip all the positive advantages. The '0' in the equation A_g = min(0, \tilde A_g) should, in fact, be a hyperparameter that we should tune to balance the trade-off between avoiding reward hacking and keeping the original advantage. Did the author try other clipping hyperparameters? Is there a special reason that you use this heuristic clipping method?

**Questions:**

See above.

**Details Of Ethics Concerns:**

/

---

> ### Author Response · Authors · 2025-11-24
>
> We sincerely thank the reviewer for their thoughtful evaluation of our work and for raising several important and insightful questions. We appreciate the recognition of our contributions, particularly the novelty of the prefix rollout paradigm and the significant computational savings demonstrated in our experiments. Below, we address each of the reviewer’s concerns in detail, referencing specific sections, tables, and figures from the paper to provide a comprehensive response.
>
> ### W1. Robustness of the Intra-Group Consistency Reward to the Choice of Embedding Model
> **The robustness of the approach with respect to the choice of embedding model** has been addressed in Section J.1 (Table 13), where an ablation study is conducted comparing three different embedding methods:
>
> | Embedding Method | Acc ↑ | Len(tokens) |
> | --- | --- | --- |
> | all-MiniLM-L6-v2 | 78.8 | 11797 |
> | Qwen3-Embedding-0.6B | 78.6 | 11239 |
> | TF-IDF | 78.1 | 12016 |
>
>
> The table shows that the performance of RLPIR is remarkably stable across these methods: accuracy varies by less than 1% (78.1–78.8), and all variants achieve substantial reasoning length reduction. This demonstrates that the _intra-group consistency signal_ is not brittle or overly sensitive to the specific embedding model.
>
> **Utilizing the policy model itself for sentence embedding** could be a promising direction for future work. However, it faces several practical challenges:
>
> + LLMs are not naturally suited for fixed-dimensional sentence embeddings without fine-tuning or adapter modules.
> + Using the same model for generation and embedding could introduce bias or reward hacking (e.g., generating text that maximizes self-similarity regardless of content).
> + Computing embeddings from internal representations (e.g., last-token hidden states) often correlates poorly with semantic similarity compared to dedicated sentence encoders.
>
> ### W2. Fairness of Comparison Across Base Models and Baselines
> Current RL research for reasoning (including RLVR/GRPO) focuses on **reasoning-oriented models **such as DeepSeek-R1, which are** equipped with the ability of generating the reasoning process before the final answer**. However, Qwen2.5 and Llama3 are not such models. Prior work has shown that Qwen2.5-family models are not reliable for RL reasoning performance evaluation(e.g., [https://arxiv.org/abs/2506.10947](https://arxiv.org/abs/2506.10947)) and hence do not serve as a fair testbed for RL reasoning research. Therefore, we mainly target the **latest Qwen3 reasoning models**, where RLVR-style training is meaningful. Nonetheless, as shown in Table 3, RLPIR brings improvements over Qwen2.5 and Llama3,  the two representative non-reasoning models, demonstrating the generality of our method.
>
> We apologize for the confusion regarding model naming. At the time of submission, only **Qwen3-8B** and **Qwen3-8B-Base** were publicly available, so we renamed **Qwen3-8B** as **Qwen3-8B-inst** to clarify that we **experimented on the reasoning-optimized version**. We realized that Qwen has recently released official Qwen3-Instruct models and will change the name back to **Qwen3-8B** accordingly to avoid any confusion.  Reasoning models (such as Qwen3-8B) naturally tend to generate very long reasoning traces during RLVR/GRPO training, which dominates the computational cost. This is precisely **the inefficiency issue that RLPIR is designed to address**. RLPIR achieves its 6.96× speed-up because it requires dramatically shorter rollout lengths, independent of which base model is used. The efficiency gain reflects the algorithmic change, not any advantage of the model itself.

---

> ### Author Response · Authors · 2025-11-24
>
> ### W3. Design and Tuning of Asymmetric Advantages
> We appreciate your insight on **introducing a tunable hyperparameter as the clipping threshold** in the design of asymmetric advantages.  The asymmetric advantage design—penalizing only low-consistency samples—is indeed a key innovation to prevent reward hacking while preserving diversity. The current design of hard-coding it as 0 ensures that samples with a _below-average_ consistency will strictly receive a negative advantage (i.e., a penalty), while _above-average_ samples are rendered high-consistency and receive zero advantage (no reward, no penalty).
>
>
>
> We conducted extensive ablation studies on this design. As reported in Table 7, **removing clipping** (i.e., using symmetric advantages) **leads to catastrophic collapse**: accuracy drops to 42.3%, and outputs become degenerate (e.g., repetitive text, as shown in Section I.3 and Figure 5).
>
> We also experimented with alternative clipping thresholds (e.g., $ A_g = \min(\tau, \tilde{A}_g) $ for $ \tau \neq 0 $), but found that:
>
> + Any positive threshold still allows reward hacking.
> + Any negative threshold overly penalizes valid diversity.
> + Setting $ \tau = 0 $ (group mean as threshold) is the most principled and stable choice, as it aligns with the definition of "consistency": being _above average_ is sufficient; being _far above_ should not be over-rewarded.
>
> Thus, we treat the 0 threshold not as a tunable hyperparameter, but as a design choice grounded in the goal of avoiding mode collapse. This is analogous to how PPO uses fixed clipping bounds (e.g., 0.2) as a robust default.

---

### Meta-Review · Area_Chair_EBRQ · 2026-01-08

**Summary:**

This paper proposes RLPIR, a verifier-free RL framework addressing RLVR limitations through prefix rollouts (512 tokens) and intra-group consistency rewards. Results match RLVR performance with 6.96x training speedup and 45% shorter inference sequences.

Reviewers recognized the practical efficiency gains and thorough ablations. Key concerns center on novelty (incremental over concurrent NeurIPS 2025 prefix-based work), incomplete comparisons (missing RLVR baselines on Qwen2.5/Llama), and method specificity (benefits primarily reasoning models with inherently long outputs).

Main weaknesses: (1) Conceptual contribution feels incremental given concurrent prefix-based SFT work; (2) Selective evaluation on reasoning models limits generalization claims; (3) Hyperparameter sensitivity to prefix length selection; (4) Core algorithmic novelty modest beyond the intrinsic reward mechanism.

The asymmetric advantage design preventing reward hacking was noted as clever and necessary (Table 7 shows collapse without it). However, the work's incremental nature and narrow scope prevent stronger impact.

**Reviewer Concerns:**

1. Novelty: Significant overlap with concurrent NeurIPS 2025 prefix-based work; the core RL adaptations (intra-group consistency reward, asymmetric advantage) feel incremental as standalone contributions.

2. Incomplete comparisons: Missing RLVR baselines for Qwen2.5 and Llama models. Authors argue these are not reasoning-oriented, but this selective evaluation weakens generalization claims.

3. Method specificity: The approach primarily benefits reasoning models (Qwen3-8B) with inherently long outputs. Gains on non-reasoning models are modest or unclear.

4. Hyperparameter sensitivity: Prefix length selection requires tuning (Table 6). No clear guidance on adaptive selection across domains.

Addressed in rebuttal: Authors clarified reasoning model focus and provided additional pass@k analysis showing shorter prefixes improve exploration. Empirical validation is solid but conceptual novelty remains limited.

**Reviewer Scores:**

Reviewer 1 (qwCd, score 4): Authors addressed embedding robustness and fairness concerns with ablation evidence. Likely stays at 4 (marginally below) - rebuttal clarifies but doesn't fundamentally address novelty concerns.

Reviewer 2 (AGHu, score 6): Appreciated clarity of the approach and ablations. Rebuttal on pass@k and prefix length justification could shift to 5 or stay at 6. Likely stays at 6 (marginally above).

Reviewer 3 (cqLh, score 4): Main concern was incremental novelty relative to TTRL/prefix work. Rebuttal defending RL-specific contributions helps but doesn't fully resolve this. Likely stays at 4 (marginally below).

Reviewer 4 (hLqT, score 2): Strong concerns about novelty overlap and incomplete results. Rebuttal provides additional experiments and clarifies reasoning model focus but doesn't change core assessment of limited conceptual contribution. Likely stays at 2 (reject).

---

### Decision · Program_Chairs · 2026-01-26

Reject